# Finite Element Simulation and Experimental Research on Uniformity Regulation of Microwave Heating of Composite Materials

**DOI:** 10.3390/polym14173484

**Published:** 2022-08-25

**Authors:** Chenglong Guan, Lihua Zhan, Shunming Yao

**Affiliations:** 1College of Mechanical and Electrical Engineering, Central South University, Changsha 410083, China; 2State Key Laboratory of High-performance Complex Manufacturing, Central South University, Changsha 410083, China

**Keywords:** finite element simulation, microwave curing, polymer-matrix composites, residual stress

## Abstract

As an attractive alternative to the traditional autoclave curing process, microwave curing has been widely used for manufacturing high-performance composites. However, the nonuniform temperature distribution during composite curing is the main problem faced by the microwave curing process, which limits its application in the aerospace industry. This paper studied the regulating effects of cavity structure and mechanical optimization methods on the uniformity of the microwave field by numerical analysis and finite element simulation, and an octagonal microwave heating device with multi-microwave generators, mode stirrers and mobile platform was developed independently and the experimental verification were finally carried out. The results showed that compared with the traditional heating device, the T800/602 carbon fiber reinforced composite laminates cured in the regulating device of microwave heating uniformity established in this paper had more uniform temperature field distribution, a more synchronous curing process and lower residual stresses.

## 1. Introduction

In order to break through the limitations of the traditional autoclave process, such as high energy consumption, low efficiency and large deformation caused by uneven temperature field and asynchronous curing, out-of-autoclave curing approaches, especially the microwave curing process, turned into an attractive alternative for curing high-performance composites [1,2]. The microwave curing process could realize the synchronous heating of the material from the inside to the outside by directly radiating the electromagnetic wave to the surface of the material, avoiding the internal temperature gradient caused by the variable size and the thick section structure of the components in traditional thermal heating [3]. At the same time, due to the characteristics of microwave selective heating, the air medium and mold in the cavity will not be heated except for the composite components, therefore it has high heating efficiency and is a new type of composite curing process with energy saving, safety and environmental protection [4]. Up to now, different kinds of studies had been conducted on microwave curing of composite materials, including fundamental principles [5,6], fiber-matrix interfaces [7,8], temperature control [9] and mechanical properties [10,11].

However, limited by the structure of the microwave heating device, the incident wave is reflected on the inner wall in the microwave cavity and then a standing wave phenomenon occurs after encountering and superimposing the reflected wave, which easily leads to the uneven distribution of the microwave field [4,12]. To date, a lot of research has been conducted in order to solve the above problems: Yi et al. used a commercial 3D finite element method to simulate the microwave distribution inside a variable frequency microwave oven. It was found that the position of the substrate could affect the harmonic frequency and field distribution [13]. Plaza-González et al. studied the effect of different mode-stirrer configurations on the electric field in a uniform microwave-heating applicator based on the finite-element method. The results showed that the stirrers’ design was critical in obtaining an acceptable uniform electric field distribution within the sample for low-permittivity materials [14]. Xu et al. discussed the uniformity of the averaged field inside a microwave oven by computer simulation and suggested that the performance of a microwave oven could be improved by introducing a stirrer [15]. Li et al. proposed a multi-pattern compensation method to achieve better uniformity of temperature on the surface of composite laminates, by monitoring the uneven temperature distribution and applying appropriate compensating heating patterns in real-time [16].

The above existing methods focused more on simulation modeling, and single regulation and analysis means, from the experimental perspective, it is difficult to provide effective guidance for the isothermal microwave heating of composite components. In this paper, through numerical calculation and finite element simulation analysis, the size of the microwave cavity was determined to take into account the heating efficiency and uniformity. Based on the mapping relationship between the electric field distribution and the temperature field distribution of the composite microwave heating process, the regulating effect of the mechanical optimization methods on the uniformity of the microwave field was systematically studied, and the design scheme of the microwave uniform heating platform for composites was formed. An octagonal microwave heating device with multi-microwave generators, mode stirrers and a load mobile platform was developed independently and experimental research was finally carried out.

## 2. Theoretical Design and Analysis of Microwave Heating Device

### 2.1. Theory and Dimension Design of Microwave Resonant Cavity

As the core component of a microwave heating system, the resonant cavity is a closed electromagnetic system completely surrounded by short-circuit surfaces or open-circuit surfaces. In an ideal microwave resonator, the cavity walls are ideal metal conductor with no dielectric loss and the cavity is filled with non-conductive medium. Frequently used microwave resonators are rectangular resonators, as shown in Figure 1; the size of the cavity is set to a×b×c (length × width × height).

According to the electromagnetic field theory, the changing electric and magnetic fields are mutually excited and connected to form a unified electromagnetic field in the resonator. As the basis for studying electromagnetic wave problems, the wave Equation (1) indicates that the energy of the electromagnetic field moves in the form of waves in the space which exists independently from electric charges and currents [4].
(1)$$\begin{array}{l} \nabla^{2}\vec{E}-\mu \varepsilon \frac{\partial^{2}\vec{E}}{\partial t^{2}}=0\\ \nabla^{2}\vec{H}-\mu \varepsilon \frac{\partial^{2}\vec{H}}{\partial t^{2}}=0 \end{array}$$
where $\vec{E}$ refers to the electric field intensity vector, ε is the dielectric constant, $\vec{H}$ is the magnetic field intensity vector and μ is the magnetic conductivity. When the field source changes time harmonically with a certain angular frequency (sine or cosine), the generated electromagnetic field also alters with the same frequency. This electromagnetic field with time-harmonic variation at a certain angular frequency is called time-harmonic electromagnetic field or sinusoidal electromagnetic field. Under certain conditions, any time-varying field can be expanded into a superposition of time-harmonic fields of different frequencies by the Fourier analysis method, and the electric and magnetic field intensity can be expressed as [17]:(2)$$\begin{array}{l} \vec{E}=\vec{E_{0}}\cos\left(\omega t+\varphi \right)\\ \vec{H}=\vec{H_{0}}\cos\left(\omega t+\varphi \right) \end{array}$$

Together with Equations (1) and (2), the wave equations can be converted into the Helmholtz equation:(3)$$\begin{array}{l} \left(\nabla^{2}+k^{2}\right)\vec{E}=0\\ \left(\nabla^{2}+k^{2}\right)\vec{H}=0 \end{array}$$
where $k=\omega \sqrt{\mu \varepsilon }$. The Helmholtz equation usually has multiple solutions, of which the most basic and simplest form is the plane wave solution. For a rectangular multi-mode resonator, there are theoretically a variety of transverse electric modes and transverse electromagnetic modes inside. Each mode represents a specific distribution form of the electromagnetic field and its specific distribution can be obtained by solving Equation (3), in which the cutoff wave number $k_{c}$ is one of the key parameters to determine the electromagnetic field distribution, its expression can be given by Equation (4) [15]:(4)$$k_{c}^{2}=k_{x}^{2}+k_{y}^{2}={\left(\frac{2\pi }{\lambda_{c}}\right)}^{2}$$
where $k_{x}$, $k_{y}$ are the standing wave numbers along the *x* and *y* axes, respectively, $k_{x}^{2}={\left(m\pi /a\right)}^{2}$, $k_{y}^{2}={\left(n\pi /b\right)}^{2}$, $\lambda_{c}$ is the cutoff wavelength, *m*, *n*, and *p* are the half wavelength numbers in the *a*, *b*, *c* directions of the cavity, correspondingly. On the other hand, since the upper and lower surfaces of the resonator are closed, the electromagnetic wave propagating inside may form standing wave nodes at the boundaries, that is, the height *c* of the cavity should be an integer multiple of the half wavelength:(5)$$c=p\frac{\lambda_{g}}{2}$$
where $\lambda_{g}$ is the guide wavelength. Generally, the cutoff wavelength $\lambda_{c}$, the guide wavelength $\lambda_{g}$ and the working wavelength λ have the following correlations [4]:(6)$$\lambda_{g}=\frac{\lambda }{\sqrt{1-{\left(\frac{\lambda }{\lambda_{c}}\right)}^{2}}}$$

Equations (4) and (5) are simply transformed respectively:(7)$$\frac{1}{\lambda_{c}^{2}}=\frac{{\left(\frac{m\pi }{a}\right)}^{2}+{\left(\frac{n\pi }{b}\right)}^{2}}{4\pi^{2}}$$
(8)$$\frac{1}{\lambda_{g}^{2}}=\frac{p^{2}}{4c^{2}}$$

Substituting Equations (7) and (8) into Equation (6), the expression of the resonant wavelength in the cavity can be obtained [4,9]:(9)$$\lambda =\frac{2}{\sqrt{{\left(\frac{m}{a}\right)}^{2}+{\left(\frac{n}{b}\right)}^{2}+{\left(\frac{p}{c}\right)}^{2}}}$$

Furthermore, the expression of the resonant frequency is given by Equation (10):(10)$$f=\frac{v}{\lambda }=\frac{C}{\lambda }=\frac{C}{2}\sqrt{{\left(\frac{m}{a}\right)}^{2}+{\left(\frac{n}{b}\right)}^{2}+{\left(\frac{p}{c}\right)}^{2}}$$
where $C=3\times 10^{8}\;m/s$ is the propagation velocity of electromagnetic waves. For TE mode, *p* cannot be zero and *m*, *n* cannot be zero at the same time; for TM mode, *p* can be zero, but neither *m* nor *n* can be zero. It can be found from Equation (10) that when the values of *m*, *n*, and *p* are different, there will be different electromagnetic field distribution modes and resonant frequencies in the cavity. The electromagnetic fields of various modes are superimposed on each other, which will form a more uniform electromagnetic field distribution in the cavity. However, the interaction of the electromagnetic waves with the metal wall of the resonator will also lead to the generation of conduction current and the appearance of power dissipation, which reduces the heating efficiency of the microwave. Therefore, it is necessary to determine a reasonable cavity size considering the conditions of the heating uniformity and heating efficiency.

In this paper, the number of resonant modes and the average electric field intensity in the cavity are selected to reflect the uniformity of the energy field distribution and the heating efficiency. The central operating frequency of the microwave generator is determined as $f_{0}=2450\;\mathrm{MHz}$; under the disturbance of the load condition, its actual operating frequency will fluctuate within a certain width, usually the frequency width Δf=50 MHz and therefore the possible resonant mode in the cavity can be calculated from Equation (11):(11)$$f_{0}-\Delta f\leq \frac{C}{2}\sqrt{{\left(\frac{m}{a}\right)}^{2}+{\left(\frac{n}{b}\right)}^{2}+{\left(\frac{p}{c}\right)}^{2}}\leq f_{0}+\Delta f$$

For different resonator sizes, the value ranges of *m*, *n*, and *p* are different and the manual calculation is inefficient, complicated and tedious. Based on the MATLAB language, a source program with high computational efficiency and reliability was written to automatically solve all the modes that might exist in the resonator. Combined with the commercial finite element software COMSOL, the electric field distribution inside the resonator under different cavity sizes was simulated. For the convenience of design and calculation, the size of the rectangular cavity was set as 1.2l×l×l(a×b×c), the generator was installed at the top of the cavity and the input power was constant at 1 kW. The *xoy* plane, *xoz* plane and *yoz* plane at the center of the cavity were selected in turn and their average electric field intensity were calculated to reflect the heating efficiency of the resonator in space. Through the above analysis and calculation, the relationship curves of the number of resonant modes and the average electric field intensity with the volume of the cavity were obtained, as shown in Figure 2.

It can be seen from Figure 2a that the number of resonant modes in the cavity changed proportionally with the increase of the size of the resonator, indicating that the number of energy field modes theoretically existing inside increased with the growth of the cavity size, which was conductive to improving the uniformity of the microwave field distribution. However, the electric field intensity inside the resonator was not a simple linear change with the rise of the cavity size, but was the result of competition between the uniformity of the energy field and the dissipation of the metal wall. When the size of the cavity was in a small range, due to the extremely non-uniformity of the field distribution, the electromagnetic waves inside were continuously reflected and superimposed to form a large number of standing waves, resulting in the general existence of areas with weak field intensity. Conversely, the dissipative effect of the small area cavity surface on the field intensity was not obvious. As the volume of the cavity increased, the uniformity of the inside energy field was gradually improved, and the average electric field intensity was significantly improved: When the cavity size increased from 0.26 m^3^ to the critical volume of 1.59 m^3^, the number of resonant modes in the cavity grew from 72 to 475 and the average field intensity raised from 2202 V/m to 28,825 V/m. Additionally, the electric field distribution and the existing resonant frequency spectrum in the cavity under the critical volume were solved, as shown in Figure 2b,c, respectively. It can be seen in Figure 2c that there were theoretically 475 resonant operating modes inside the cavity under the critical dimension, which met the requirement that a typical uniformly microwave heating resonator should have at least 108 operating modes [15]. In the meantime, for the multi-mode resonator with the center frequency of 2450 MHz, it can be considered that the cavity had a relatively uniform electromagnetic field distribution because the spread of the resonant frequencies corresponding to different modes in the full frequency width was relatively consistent and there was a similar number of resonant frequencies located on both sides of the center frequency [18,19,20].

When the dimension of the resonator continued to increase beyond the critical volume, the energy field inside the cavity had formed a fairly uniform distribution state. At this time, the energy consumptions generated by the large area metal surfaces played a major role in the attenuation of the electric field intensity, which was manifested as a gradual decrease in the electric field intensity with the increase of the resonator size. Nevertheless, as the volume of the cavity further increased, the slope of the electric field intensity curve gradually became slower and tended to be stable, which meant that the energy dissipation created by the metal surfaces of the cavity was limited on the premise of the relatively uniform energy field inside. Meanwhile, a certain power dissipation brought by the metal cavity could effectively prevent the heating platform from being damaged due to excessive energy field intensity when the platform was not loaded. In conclusion, under the premise of taking into account the heating uniformity and efficiency, the size of the microwave resonator should be at least greater than the critical volume of 1.59 m^3^, which could have a high electric field intensity while ensuring the existence of more resonant modes inside the cavity.

### 2.2. Finite Element Simulation on Microwave Field Uniformity Optimization

#### 2.2.1. The Basic Theory of Physics for Microwave Heating

The power flow through a closed surface can be calculated by integrating the Poynting vector in complex form, as shown in Equation (12) [17]:(12)∮SP⇀·dS⇀=Re∮(E⇀×H⇀*)2dS⇀

The expression of the medium average heat loss can be obtained by integrating the differential form of the Poynting theorem and combining it with Equation (12) [17]:(13)Pav=−12∮SRe(E⇀×H⇀*)·dS

In a time-harmonic electromagnetic field, the differential equation of the Maxwell– Ampere law can be written in complex form:(14)$$\nabla \times \vec{H}=\vec{J}+j\omega \varepsilon_{0}\varepsilon^{*}\vec{E}$$
where $\vec{J}$ is the current density vector, $\varepsilon_{0}$ and $\varepsilon^{*}$ represent the vacuum dielectric constant and complex dielectric constant, respectively. Substituting $\vec{J}=\sigma \vec{E}$ and $\varepsilon^{*}=\varepsilon^{\prime }-j\varepsilon^{\prime \prime }$ into Equation (14):(15)$$\begin{array}{cc} \nabla \times \vec{H} & =\sigma \vec{E}+(\omega \varepsilon_{0}\varepsilon^{\prime \prime }+j\omega \varepsilon_{0}\varepsilon^{\prime })\vec{E}\\ & =\omega \varepsilon_{0}\varepsilon_{eff}\vec{E}+j\omega \varepsilon_{0}\varepsilon^{\prime }\vec{E} \end{array}$$
where $\varepsilon_{eff}=\varepsilon^{\prime \prime }+\frac{\sigma }{\omega \varepsilon_{0}}$ is the effective loss factor, σ is the electrical conductivity, $\varepsilon^{\prime }$ is the dielectric constant and $\varepsilon^{\prime \prime }$ is the dissipation factor. Transforming both sides of Equation (15) and multiply the electric field intensity vector $\vec{E}$:(16)$$(\nabla \times {\vec{H}}^{*})\cdot \vec{E}=\omega \varepsilon_{0}\varepsilon_{eff}{\vec{E}}^{*}\cdot \vec{E}-j\omega \varepsilon_{0}\varepsilon^{\prime }{\vec{E}}^{*}\cdot \vec{E}$$

Similarly, the differential equation of Faraday’s law of electromagnetic induction is written as the complex form and the conjugate complex vector ${\vec{H}}^{*}$ is multiplied at both ends of the equation:(17)$$(\nabla \times \vec{E})\cdot {\vec{H}}^{*}=-j\omega \mu_{0}\mu^{\prime }\vec{H}\cdot {\vec{H}}^{*}$$
where $\mu_{0}$ and $\mu^{\prime }$ represent the vacuum magnetic permeability and the medium magnetic permeability, respectively. Subtracting Equation (16) from Equation (17):(18)$$\begin{array}{cc} (\nabla \times \vec{E})\cdot {\vec{H}}^{*}-(\nabla \times {\vec{H}}^{*})\cdot \vec{E} & =-j\omega \mu_{0}\mu^{\prime }\vec{H}\cdot {\vec{H}}^{*}-\omega \varepsilon_{0}\varepsilon_{eff}{\vec{E}}^{*}\cdot \vec{E}\\ & +j\omega \varepsilon_{0}\varepsilon^{\prime }{\vec{E}}^{*}\cdot \vec{E} \end{array}$$

Integrating both ends of Equation (18) and combining the divergence theorem [17]:(19)$$\begin{array}{cc} \oint_{V}\nabla \cdot (\vec{E}\times {\vec{H}}^{*})dV=\oint_{S}(\vec{E}\times {\vec{H}}^{*})\cdot dS & =-j\omega \oint_{V}\left(\mu_{0}\mu^{\prime }\vec{H}\cdot {\vec{H}}^{*}-\varepsilon_{0}\varepsilon^{\prime }\vec{E}\cdot {\vec{E}}^{*}\right)dV\\ & -\oint_{V}\omega \varepsilon_{0}\varepsilon_{eff}\vec{E}\cdot {\vec{E}}^{*}dV \end{array}$$

Substituting Equation (19) into Equation (13) [17]:(20)$$P_{av}=\frac{1}{2}\omega \varepsilon_{0}\varepsilon_{eff}\oint_{V}(\vec{E}\cdot {\vec{E}}^{*})dV$$

When the electric field intensity vector is a constant, Equation (20) can be converted to:(21)$$P_{av}=\omega \varepsilon_{0}\varepsilon_{eff}E_{rms}^{2}V$$
where $E_{rms}$ is the effective value of the electric field intensity, *V* is the volume of the medium to be heated.

The heating rate of the composite material while absorbing microwave energy can be given by Equation (22):(22)$$P=\frac{Q_{t}}{t}=mc\cdot \left(\frac{T-T_{0}}{t}\right)$$

Combining Equations (21) and (22), the expression of the heating rate of the composite material in the microwave field can be obtained:(23)$$\frac{T-T_{0}}{t}=\frac{\omega \varepsilon_{0}\varepsilon_{eff}E_{rms}^{2}}{\rho c}$$
where ρ is the density and *c* is the heat capacity of the composite material.

It can be seen from Equation (23) that for a microwave resonator with the constant central frequency, the heating rates of the laminates are only proportional to the square of the electric field intensity in the resonator and the electric field distribution in the cavity can be fully mapped to the temperature field situation of composite laminates under microwave heating conditions.

According to the previous analysis, for the microwave resonator studied in this paper, there are theoretically multiple resonant modes in the frequency range of 2400 MHz to 2500 MHz. In fact, restricted by the shape of the resonator, the number and the location of microwave generators, a considerable part of modes will not be motivated, which will affect the uniformity of the microwave field inside the cavity. Therefore, in order to achieve synchronous heating of composite components with different shapes and sizes during the microwave curing process, it is necessary to further analyze and optimize the distribution of the electric field in the resonator.

#### 2.2.2. Establishment of the Finite Element Model

Based on the advantages of simple application modes, visible simulation processes and a rich model library, COMSOL Multiphysics was selected to analyze and optimize the electric field uniformity in microwave resonator. Combined with the previous research results and the basic experimental requirements, the size of the resonator was determined as $1300\;\mathrm{mm}\;\times 1100\;\mathrm{mm}\;\times 1100\;\mathrm{mm}\;\left(a\times b\times c\right)$ [21,22]. The octagonal resonator shape, which was the mainstream in the existing research, was selected, and the software SOLIDWORKS was used for geometric modeling. Meanwhile, considering the influence of the number and the location of microwave generators on the uniformity of the energy field, an independent heating source was set on each surface of the cavity and its specific number was shown in Figure 3a.

On this basis, a total of 8 sets of mode stirrers were installed at the incidence position of each generator to further explore the regulation of the mode stirrer on the electric field distribution. Through the geometry-import operation, the geometry model established was directly imported into the COMSOL, the material of the cavity wall and the mode stirrers was set to stainless steel 302, and the loading platform was a SiO_2_ plate. The microwave heating module was added as the physical energy field to be analyzed, the Helmholtz Equation (3) was used as the governing equation and the mesh type was selected as tetrahedral mesh, as shown in Figure 3b. A steady-state solver was chosen for frequency domain analysis and the microwave frequency was set to 2.45 GHz.

#### 2.2.3. Design and Selection of Microwave Generator

##### Influence of Microwave Generator Size on Electric Field Distribution

The selection of the width and height of the microwave generators played a decisive role in the cutoff wavelength of the microwave that could propagate in it. For the microwave with a frequency of 2.45 GHz (wavelength of about 122.4 mm) used in this paper, if it was required to operate in the waveguide, the wider side of the waveguide should be at least greater than the half-wavelength, namely 61.2 mm. According to the standard waveguide size specified in Chinese standard GB/T 11450-1989 and comprehensively considering the manufacturing cost, the BJ-32 waveguide with a width of 72.14 mm and a height of 34.04 mm was selected, which could achieve the maximum transmission power under the premise of ensuring the frequency bandwidth.

In actual conditions, there was a need to reserve space (about 250 mm) on the side near the door of the cavity for the installation of the observation window, microwave shielding strips and other structures, and it was required that holes be opened on the surface of the cavity to facilitate the introduction of vacuum tubes, temperature measuring fibers and other lines. Therefore, the maximum length of the microwave generator should not exceed 1000 mm. Combined with the width and height determined above, the lengths of the generator were selected to be 400 mm, 800 mm and 1000 mm, respectively, and the assembly position was the axial center of the resonator, as shown in Figure 4. The total microwave power under different conditions was set to be 1 kW constantly, and the influence of the length of the microwave generator on the electric field was explored.

Figure 5a shows the electric field nephograms of the loading platform with different microwave generator lengths, in which red represented the region with higher electric field intensity while blue indicated the position with lower field strength. It can be clearly observed from the figure that when the length of the resonator was 400 mm, the field distribution on the loading platform was extremely uneven and there was a large area with high electric field strength, mainly located in the range of the microwave generator radiation. By contrast, the field intensity was significantly lower in the area not covered by the microwave generator. When the length of the generator increased to 800 mm, the uniformity of the electric field was improved to a certain extent. Although there were still some locations with higher field strength, the phenomenon of wide areas with low field intensity was well ameliorated. As the length of the generator became 1000 mm, the overall electric field on the platform became more even. Therefore, with the improvement of the length of the microwave generator, since the area where the microwave could be imported increased gradually, the range covered by the microwave constantly expanded and the electric field distribution on the platform tended to be homogeneous. In order to quantitatively describe the electric field location in the resonator, the coefficient of variation (CoV) was introduced in this paper, which is defined by Equation (24) [21]:(24)$$\bar{\sigma }=\frac{\sqrt{\frac{\sum_{i=1}^{n}{\left(\left|E_{i}\right|-\left|\bar{E}\right|\right)}^{2}}{n}}}{\left|\bar{E}\right|}$$
where $\bar{\sigma }$ is the coefficient of variation, *n* is the number of statistical positions, $\left|E_{i}\right|$ is the absolute value of the electric field strength at each position and $\left|\bar{E}\right|$ is the mean value across all locations. The smaller the value of the CoV, the better the uniformity of the electric field distribution. By calculating the CoV of the electric field intensity on the platform under different microwave generator length conditions, it could be found that when the length increased from 400 mm to 1000 mm, the CoV of the electric field strength on the platform was 0.378, 0.357 and 0.328, respectively, which proves that the field distribution on the platform gradually improved with the raise of the length of the generator.

Due to the complex spatial geometry of aerospace composite material components, the spatial electric field distribution in the cavity also had a significant impact on the overall heating uniformity. Therefore, the influence of the generator length on the uniformity of the spatial electric field in the resonator should be further clarified. The nephogram of the spatial electric field inside the resonator under different conditions was drawn, as shown in Figure 5b, and it can be seen from the figure that when the length of the generator increased from 400 mm to 1000 mm, the overall spatial electric field in the cavity also tended to be uniform, and the problem of lower local field intensity in the cavity caused by the smaller import area was well solved. In conclusion, for the microwave resonator with a certain dimension in this paper, it was advisable to select the microwave generator size with a length, width and height of 1000 mm, 72.14 mm and 34.04 mm, respectively, which could obtain a relatively uniform electric field distribution inside the microwave resonator.

##### Influence of the Microwave Generators Quantity and Setting Modes on the Electric Field

On the basis of the selected size of the microwave generators, the effect of their quantity and setting modes on the electric field distribution in the resonator were further analyzed. The number of the working generators was set from 1 to 8, and the simulation was carried out in turn. Different working positions with the same quantity of generators were also considered and the specific setting methods were shown in Table 1, which mostly covered the possible working forms of the microwave generators. The working positions of the generators were all numbered, as shown in Figure 3. For the sake of variable control, the total microwave power fed into the resonator was the same under different working conditions (set to 1 kW and were evenly distributed to each generator). Similarly, the CoV was calculated based on Equation (24) as an index to measure the uniformity of the electric field.

The model established in Section 2.2.2 was used to carry out simulation analysis on the different setting methods of microwave generators listed in Table 1, and the CoV of the electric field intensity on the loading platform was obtained, as shown in Table 2. It can be seen that the position of the in-plane electric field basically tended to be uniform with the increase of the number of microwave generators, and was also closely related to the locations of the generators. According to three sets of the conditions numbered [(9),(10)], [(19),(20)] and [(23),(24)], it was oserved that when the microwave generators located under the resonator worked, the in-plane electric field were more even. The setting methods numbered [(7),(16)], [(9),(17)] and [(10),(18)] showed that the activation of two symmetrical generators on the left and right sides of the octagonal resonator (i.e., ③ and ⑦) would be more conducive to the uniformity of the electric field. Comparing the setting plans numbered [(11),(13),(14),(23),(24)] with other forms, the homogeneity of the in-plane electric field was significantly improved when the microwave generators were symmetrically distributed along the cross-section of the cavity. Combined with the above statistical analysis results, the electric field uniformity on the loading platform was the best when the microwave generators worked with the number (23) type (CoV = 0.304) in Table 1, and the homogeneity of the in-plane electric field increased by 25.3% compared with the unreasonable number (6) working mode (CoV = 0.407).

In order to visually show the variation rule of the electric field distribution in the cavity, four representative microwave generator working modes in Table 1 were selected: number (6), number (9), number (16) and number (23), which basically covered the range of all the coefficients of variation in Table 2. The electric field nephograms on the loading platform of the above four working conditions were drawn, as shown in Figure 6. It could be clearly observed form the figures that as the number of microwave generators increased and the arrangement tended to be symmetrical along the cavity, the ‘hot spots’ and ‘cold spots’ with excessively high or low electric field strength gradually decreased, and the uniformity of the in-plane electric field distribution progressively improved.

Furthermore, the homogeneity of the electric field on the loading platform under different microwave generator setting conditions was quantitatively analyzed. The associated Cartesian coordinate system on the loading platform was established, as shown in Figure 7a, and the coordinates along the radial direction (*x* direction) of the resonator were selected as *x* = −300 mm and *x* = 0, *x* = 300 mm to map the whole loading platform. The variation curves of the electric field intensity along the axial direction (*z* direction) of the cavity were separately drawn, as shown in Figure 7b. It can be seen from the figure that there were significant differences in the distribution of the in-plane electric field under different numbers and setting conditions of microwave generators. When the generators worked as an asymmetric type of number (6), the intensity of the electric field fluctuated quite violently, the maximum field strength was close to 50,000 V/m, about 10 times the minimum value, and the overall distribution was extremely uneven. When the generators were arranged in the way of number (9), the maximum field strength was about 32,792 V/m and the minimum field intensity was about 5072 V/m. The overall uniformity of the electric field was evidently enhanced, especially in terms of the high field strength caused by the concentrated distribution of the electric field. With the further increase of the number of microwave generators and the arrangement of symmetrical distribution along the cavity, the fluctuation of the electric field strength at different positions along the axis of the resonator gradually decreased, the points with exceedingly high or low field strength disappeared, and the distribution of the in-plane electric field tended to be even.

To find out whether the influence of the number and the arrangement of microwave generators on the electric field distribution was applicable to the spatial electric field in the resonator, so as to realize the uniform and synchronous heating of aerospace composite components with complex geometrical features, the uniformity of spatial electric field distribution was studied on the basis of the above analysis. Four typical microwave setting forms in Table 1 were selected and the spatial electric field nephograms inside the resonator under different conditions were drawn, as shown in Figure 8. Referring to the coordinate system established in Figure 7a, electric field cross-sections along the axial direction of 100 mm, 300 mm, 500 mm and 700 mm in the resonator were extracted, and the distribution of spatial electric field under different setting forms was directly compared (Figure 9).

It can be seen from Figure 8 and Figure 9 that with the changes in number and the locations of microwave generators, the spatial electric field distribution in the resonator still varied greatly. When microwave generators were arranged in the number (6) way, the overall distribution of the spatial electric field inside the cavity was extremely uneven, and there were multiple areas with a concentrated electric field or incomplete coverage in each section and even the entire space, which greatly reduced the uniformity of temperature field during the composites curing process, resulting in poor heating efficiency and high manufacturing cost. With the increase in the quantity of microwave generators and their tendency to be symmetrically located along the cavity, the spatial electric field in the resonator also presented a trend of uniform distribution, and the coexistence of local stronger and lower fields in each transverse section gradually disappeared, indicating that the adjustment of the working number and the setting method of microwave generators had a significant effect on improving the spatial distribution of the electric field in the resonator.

According to the above qualitative and quantitative analysis, it could be seen that, for the microwave heating device with a certain size studied in this paper, the spatial and the in-plane electric field distribution inside the resonator could be significantly optimized by adjusting the working quantity and setting form of the microwave generators. At the same time, by selecting the microwave generators working mode with the serial number (23) in Table 1, a more ideal spatial electric field could be obtained inside the resonator, which could play an important role in realizing the uniformity and synchronous microwave heating of composite laminates in the following experimental verification.

#### 2.2.4. Design and Selection of Microwave Mode Stirrer

##### Size Design of Microwave Mode Stirrer

The mode stirrer refers to a mechanical structure which can rotate periodically inside the microwave resonator, reflecting the microwaves in all directions to achieve a more even field distribution and a better heating effect [23]. The microwave mode stirrer is usually made of a conductive material such as metal, and its geometric parameters are closely related to the electric field distribution in the resonator. In order to better stir and reflect the microwaves fed by the generators, the length of the mode stirrers should be as consistent as possible with the generators, and should be an integer multiple of the wavelength. Considering the structure of the door and the operability of the composite components curing process, the length of the stirrers should be slightly shorter than the microwave generators. Therefore, the length of the mode stirrers was selected to be 7 times the wavelength, namely 854 mm.

In terms of the width selection of the mode stirrers, it can be found from Figure 9 that due to the influence of the metal material of the mode stirrers, the phenomenon of tip arc discharge was inescapable when they worked, and this was likely to cause damage to parts in the cavity and even to the microwave generators. Meanwhile, considering that the rotation radius of the stirrers might affect the maximum molding size of the composite components, the width of the mode stirrers should be as small as possible. However, when the width of the stirrers was too small, especially less than the wavelength (122 mm in this article), the metal stirrers failed to reflect the microwaves, but travelled in the form of surface waves on the metal boundaries [24,25]. In summary, the width of the microwave stirrers should be as small as possible on the premise of greater than the wavelength. In this paper, 150 mm was selected as the width of the microwave mode stirrers.

##### Influence of the Periodic Rotation of Mode Stirrers on the Electric Field

In addition to its own structural parameters, the angle between the mode stirrer and the generator is also critical to the spatial electric field distribution in the microwave cavity [16,24]. Two types of microwave generators working modes with relatively uniform and nonuniform initial electric field, namely number (6) and number (23) in Table 1, were chosen and the electric field distribution during the rotating process was extracted and compared. The angle between the microwave mode stirrer and the surface of the resonator was defined as α, as shown in Figure 7a, and α=0°, 45°, 90°, 135° were chosen in turn.

Figure 10 showed the nephograms of the electric field on the loading platform when the mode stirrers rotated to different positions under two types of generators’ working conditions. It can be seen from the figures that for different microwave generators’ working forms, the corresponding electric field distribution in the cavity was not constant during the mode stirrers’ rotating process, and the regions with relatively higher or lower electric field strength alternated continuously as the angles between the mode stirrers and the generators changed. Therefore, after introducing the mode stirrers into the microwave resonator, the composite components placed on the platform would be heated by the electric field with periodic strong and weak alternating changes, avoiding the uneven temperature field caused by the heterogeneous initial electric field during the composites curing process and further improved the uniformity of microwave heating.

In order to quantitatively investigate the effect of the periodic rotation of the mode stirrers on the electric field, the electric field strength–displacement curves on the loading platform were drawn when the mode stirrers rotated to certain positions. Referring to the coordinate system established in Figure 7a, the variations of the electric field intensity along the axis (*z* direction) at the radial direction *x* = 0 of the cavity were extracted, as shown in Figure 11. It could be found that under different working types of microwave generators, the electric field in the resonator would follow the rotation of the mode stirrers to generate real-time changes. For example, under the number (6) arrangement of microwave generators, the in-plane electric field strength at the coordinate *z* = 43.5 mm was 4702 V/m, 7982 V/m, 11,741 V/m and 43,991 V/m in turn with the rotation of the mode stirrers. Under the generators arrangement of number (23), the electric field intensity at the coordinate *z* = 451.6 mm on the platform presented 6110 V/m, 18,795 V/m, 7982 V/m and 3526 V/m. The electric field intensity at the same position changed periodically and alternately, which solved the problem of uneven distribution of the initial electric field.

The average electric field intensity in one rotation cycle of the mode stirrers was calculated and compared with the initial electric field, before introducing the mode stirrers into the microwave resonator as shown in Figure 12. As can be seen from the figures, the periodic rotation of the microwave mode stirrers in the resonator well solved the problem of uneven initial electric field distribution caused by the unreasonable setting of generators. The maximum field strength at the axis of the loading platform was 42,193 V/m, the minimum field strength was 5138 V/m, and the maximum difference value was up to 37,055 V/m. After the introduction of the mode stirrers, the maximum value of the average field strength was 18,166 V/m, the minimum value was 4721 V/m, and the maximum difference was 13,445 V/m. Compared with the case where the mode stirrers were not introduced, the maximum difference of the field strength dropped by as much as 63.7%. On the other hand, under the reasonable number (23) generators arrangement, the introduction of the mode stirrers could further optimize and improve the electric field intensity distribution. Compared with the maximum field strength difference of 13,610 V/m, the periodic rotation of the mode stirrer could reduce the maximum difference of the average field strength to 6108 V/m, a decrease of 55.1%.

#### 2.2.5. Design and Selection of Loading Platform

In order to make the microwave generators located on the bottom of the octagonal resonator heat the composite components placed on the loading platform better, the material of the loading platform should be microwave-penetrating material. For the comprehensive consideration of durability and manufacturing economy, heat-resistant ceramics were finally selected as the material of the loading platform. Since the loading platform in the resonator would not affect the distribution of the electric field itself, referring to the design of the household microwave oven, the relative motion of the loading platform was generated to allow the composite components to be heated alternately between the regions with different field strengths; finally, the effect of uniform heating was achieved. However, due to the complicated arrangement of temperature measuring fibers and vacuum pipelines in the resonator, the common disc-rotating moving scheme could not be applied to the actual microwave heating process of composite components. Therefore, for the microwave heating device designed in this paper, the moving mode of the loading platform was selected as translational type, which meant the platform could move in a straight line along the axis of the cavity. Considering that the longer the moving distance, the smaller the size of the loading platform, the maximum molding size of composite components was limited and could not meet the needs of basic experiments. Based on the above analysis, the maximum displacement of the loading platform was chosen to be 200 mm and the maximum size was designed to be 1100 mm × 800 mm.

## 3. Materials and Methods

### 3.1. Regulating Device of Microwave Heating Uniformity

Based on the calculation and simulation results mentioned above, combined with the basic experimental requirements of this paper, a regulating device of microwave heating uniformity was independently designed and developed. The cross-section of the resonator was a regular octagon, the diameter of the circumcircle was 1100 mm and the length of the device was 1300 mm, as shown in Figure 13. Each surface of the cavity was equipped with a microwave generator along the length direction of the resonator, the length of the generator was 1000 mm and the microwave frequency was 2.45 GHz. The maximum output power of a single generator was 1000 W and it could be continuously adjusted in the range of 100–1000 W. The maximum total output power of the device was 8000 W.

A matching mode stirrer was installed at the output port of each generator and the microwave’s input was continuously reflected and superimposed by periodic rotation along the length of the cavity. The stirrers were made of 304 stainless steel, with a length of 854 mm and a width of 150 mm. A stepper motor (model: LEADSHINE Control Technology Company, 42HS02, Shenzhen, China) was assembled behind the cavity and drove the mode stirrers to rotate at a constant speed by chain transmission, the speed was adjustable in series within the range of 1–10 r/min. Another stepper motor (model: LEADSHINE Control Technology Company, 86HS120, Shenzhen, China) was used in conjunction with the gear-rack mechanism to create the reciprocating motion of the loading platform along the axial direction in the cavity. The moving distance was adjustable within the range of 50–200 mm and the speed was a constant 50 mm/min. The heat-resistant ceramics were selected as the material of the loading platform, with a maximum carrying capacity of 250 kg.

### 3.2. Materials and Devices

The carbon fiber reinforced epoxy prepreg (T800/602, fiber volume friction of 65%) was employed in this experiment, provided by Aerospace Long March Arimt Technology Co., Ltd. of China (Beijing, China). After 24 h thawing treatment at 22 °C constant temperature, the composite laminates with the ply sequence of [0/90/0/90/0/90/0]_s_ were prepared and each size was 300 mm × 300 mm (length × width). Four temperature measuring fibers (model: Optsensor, HQ-FTS-PEB0B-0300, Xi’an, China) were placed between the 7th and 8th layers pf the laminates to realize the real-time monitoring and feedback during the microwave heating process, as shown in Figure 14a. A composite tool made of the same material system was used and the aluminum tapes were stuck on the edge of the tool to suppress the discharge breakdown of carbon fibers.

The laminates were placed into the regulating device of microwave heating uniformity established in this paper and the traditional rectangular heating device with approximately equal size (as shown in Figure 14b) respectively, then heated from room temperature to 130 °C at a heating rate of 2 °C/min for 120 min. The generators of the regulating device were set according to the type of number (23) in Table 1 and the rotation rate was set to 6 r/min. Vacuum treatment was applied to all laminates during the whole curing process.

### 3.3. Test and Equipment for the Curing Degrees of Resins

Differential scanning calorimetry can measure the temperature of the curing reaction of the resins at a certain heating rate and the energy released during the curing process, and is an effective method to quantitatively test the degrees of the resins [26]. To clarify the influence of temperature field distribution on the curing degrees of resins during microwave curing, a differential scanning calorimeter (model: PerkinElmer, DSC 8500, Waltham, MA, USA) was used in this paper to test the curing degrees in different areas of the microwave cured laminates. According to the Chinese standard HB 7614-1998, the total reaction heat of the prepreg and the curing residual reaction heat of the specimens sampled from different areas of the laminates were measured, respectively. The test temperature ranged from 30 °C to 300 °C, with a heating rate of 10 °C/min. High purity helium gas was used as protective gas, with a flow rate of 20 mL/min. After the tests, the curing degrees of the resins were calculated by Equation (25) [27]:(25)$$\alpha =(H_{total}-H_{r})/H_{total}$$
where α is the curing degree of the resins, $H_{total}$ is the total reaction heat of the prepreg and $H_{r}$ is the residual reaction heat of the specimens.

### 3.4. Test and Equipment for the Stresses/Strains during the Microwave Curing Process

When the fiber Bragg grating (FBG) sensors were used to monitor the curing process of composite laminates, the strains acting on the grating and the changes of the nearby temperature field would cause the varieties of the effective refractive index of the grating, finally reflected as the drift of the central reflection wavelength. Therefore, the strains acting on the grating could be calculated by the changes in temperature and the central reflection wavelength, as shown in Equation (26) [28]:(26)$$\Delta \lambda_{B}=K_{\varepsilon }\cdot \varepsilon +K_{T}\cdot \Delta T$$
where $\Delta \lambda_{B}$ is the fluctuation of the reflection wavelength, ε is the axial strain of the fiber, ΔT is the temperature variation near the grating, $K_{\varepsilon }=$ 0.0012 nm/με is the sensitivity coefficient of strain and $K_{T}=$ 0.0095 nm/°C is the sensitivity coefficient of temperature. The temperature measuring fibers were also used to solve the issue of FBG sensors not collecting the temperature variation while recording the fluctuation of the central reflection wavelength.

In order to explore the influence of the temperature field on the composites strain during the curing process, two FBG sensors as well as two temperature measuring fibers were embedded in the areas with the largest temperature difference between the 7th and 8th layers of the laminates. All FBG sensors were placed perpendicular to the carbon fibers’ orientation, as shown in Figure 15a. The distance between the temperature measuring fiber and the corresponding FBG sensor was controlled at 5–20 mm, so that the temperature collected by the temperature measuring fiber could be used to replace the temperature change of the FBG sensor. The strain monitoring process during microwave heating of composite laminates was shown in Figure 15b.

Moreover, a composites equivalent modulus calculation software combined with the MATLAB programming language was developed in this paper. By inputting the mechanical property parameters and ply sequence of the composite laminate, the equivalent modulus of the laminate under specific ply conditions was obtained, as shown in Figure 15c. Since the composite laminates were completely cured after the heat preservation stage, the effect of the temperature on the equivalent modulus of the laminates in the cooling stage was ignored and the equivalent modulus was assumed to be a constant value.

## 4. Results and Discussion

### 4.1. Verification of Temperature Field Uniformity during Microwave Curing Process

The internal temperature changes of the composite laminates placed in the temperature field regulating device and the traditional heating device during the curing process were monitored and collected, as shown in Figure 16. It can be seen from the Figure 16a that the internal temperature field of the laminates inside the traditional heating device was relatively uneven. With the continuous increase of the curing temperature, the temperature difference between various measuring positions gradually increased, and reached the maximum value of 17.4 °C at the end of the heating stage. After entering the insulation stage, the temperature difference between various measurement areas inside the laminates was maintained at about 10 °C and the minimum temperature was only 119.5 °C, which far from met the uniformity of temperature distribution ≤±5 °C required by the curing process. The fundamental reason for the above phenomenon was that under the traditional microwave heating method, the electric field in the rectangular resonator was extremely uneven. The areas of laminates with high field intensity responded quickly and heated up under the excitation of microwave energy, while the areas with lower field strength could not get the same power input, leading to the accumulation and continuous growth of the internal temperature difference during the heating stage.

On the other hand, the internal temperature field distribution of the composite laminates cured in the regulating device was relatively uniform, as shown in Figure 16b. The temperature changes in different measuring areas inside the laminates were almost the same, the maximum temperature difference was only 2.8 °C and the minimum value was 0 °C. After entering the insulation stage, the temperature difference between various measuring areas inside the laminates was controlled within 5 °C, corresponding with the lowest internal temperature of 126.4 °C. At the end of the insulation stage, the internal temperatures of different measuring positions were 126.5 °C, 129.9 °C, 129.1 °C and 127.6 °C, respectively; the overall temperature field distribution was ideal. Benefiting from the significant improvement of the electric field distribution by the actions of various regulating methods in the octagonal resonator, the uniformity of the internal temperature field distribution of the laminates cured in the regulating device was significantly improved during the heating process; the maximum temperature difference inside the laminates dropped from 17.4 °C to 3.6 °C, a decrease of 79.31%. Meanwhile, the temperature difference between various areas inside the laminates was controlled within ±5 °C, which fully met the temperature uniformity requirements of the T800/602 aerospace composites system.

### 4.2. Effect of Uniformity of Temperature Field on Curing Degree of Composites

The prepregs and microwave cured laminates were sampled and subjected to scanning calorimetry analysis, and the heat flow–time–temperature curves of the cured samples and the prepregs are shown in Figure 17. As can be seen from Figure 17a, the DSC curve of the uncured prepregs had an obvious curing exothermic peak. Integrated, the exothermic peak with time and the peak area obtained was the total heat of reaction of the resins. For the samples cured by microwave in Figure 17b, although the DSC curves no longer had a significant reaction exothermic peak, the curing degrees of the specimens in different positions were not identical, due to the difference in the internal temperature field of the composite laminates during the heating process. For the incompletely cured sample, the DSC curve still had a small exothermic peak within the same time and temperature range. Integrated, the exothermic peak with time and the peak area obtained was the residual reaction heat of resins. For the fully cured sample, there was no obvious exothermic peak in the DSC curve, which means that the residual reaction heat of resins was extremely small.

Combined with the DSC curves and Equation (25), the curing degree of each sample was calculated and shown in Table 3. For composite laminates cured in the traditional heating device, due to the maximum temperature difference of up to 17.4 °C between the various positions inside the laminates during the heating process, the curing degrees of specimens sampled from the laminates were quite different. The residual reaction heat of the sample with the lowest curing degree was 4.45 J/g, which was 34.2 times that of the sample with the highest curing degree, and the difference between the two was 5.59%. In engineering applications, the curing degree of the resin matrix was usually required to reach about 99% [29], but due to the asynchronous heating caused by the uneven distribution of the electric field, part of the laminates could not reach the curing temperature required by the process during the microwave heating, resulting in asynchronous curing of the resins in different areas.

On the other hand, for the composite laminates cured in the temperature field regulating device, benefiting from the uniform distribution of the temperature field during the heating process, the temperature changes in different areas of the laminates were highly consistent and the curing process of resins was carried out synchronously. When the curing process finished, the curing degrees obtained by specimens sampled from different positions of the laminates were basically the same, all reaching more than 99% and considered to be completely cured. The residual reaction heat of the sample with the lowest curing degree was only 4.5 times that of the sample with the highest curing degree, and the difference between the two was only 0.77%; the overall curing synchronization was significantly improved.

### 4.3. Effect of Uniformity of Temperature Field on Strains/Stresses of Composites

Online monitoring and acquisition of strains inside the laminates cured under two different heating conditions were carried out and the results are shown in Figure 18. It can be seen from the figures that the internal strains of the composite laminates showed the same change rule. In the heating stage, the viscosity of the resins gradually decreased with the growth of the temperature, the inhibitory effect of the fibers on the resins was weakened and the tensile strains of the sensors increased, which was intuitively manifested by the rapid rise of the curve and the incline of the slope. When the laminates entered the initial stage of heat preservation, the thermal expansion effect of the resins was weakened because of the constant ambient temperature. At this time, the curing reaction rate of the resins was significantly increased and curing shrinkage of the resins occurred, resulting in a decrease of the laminates’ internal tensile strains. The third stage was the mid-late insulation stage of the laminates, the cross-linking reaction of the resins was basically complete, the matrix did not continue to shrink and the internal strains of the laminates remained constant. During the cooling stage, the internal strains of the laminates decreased rapidly due to the further shrinkage of the matrix with the decrease of the ambient temperature.

Comparing the monitoring curves inside the laminates under different heating conditions, it can be found that under the condition of a relatively uniform temperature field, the strain changes in different measuring areas of the laminates were basically the same, as shown in Figure 18a. At the beginning of the insulation stage, the two measuring points both reached the maximum strains, which were 273.01 µε and 242.82 µε, respectively. In the heat preservation stage, since the internal temperature field of the laminates was relatively uniform, the resins could be cured simultaneously and the degree of curing shrinkage of the resins in each area were basically the same; the strains at different measuring points in this stage were stable at about 180.71 µε and 165.18 µε, respectively. Due to the maximum temperature difference inside the laminates being only 3.1 °C at the end of the heat preservation stage, the curves of the two measuring points in the cooling stage almost coincided. Until the experiments finished, the curing residual strains of the measuring points were −255.84 µε and −251.69 µε. For the laminates placed in the traditional heating device, although the changes of the internal strains at each measuring point still presented the same trend, there was a significant difference in the values. Since the thermal expansion of the resins caused by the temperature change had a great influence on the strains, the value corresponding to the measuring point 2 with the high temperature was higher at the initial insulation stage and reached 326.92 µε. The strain value of measuring point 1 with a temperature difference of 17.4 °C from point 2 was only 251.97 µε, and the maximum strain difference of various measuring points was 22.93%. At the end of the heat preservation stage, it can be seen from Figure 16a that there was still a temperature difference of 8.8 °C between the two measuring points. Therefore, compared with the measuring point 1 with a lower temperature, the shrinkage of the resins had a more significant effect on the strains at the measuring point 2, with a higher temperature. The curing residual strains of the two measuring points after the experiment were −246.87 µε and −345.69 µε, respectively. The strains inside the composite laminates under the two microwave heating conditions were statistically compared, as shown in Table 4.

On the basis of the above analysis, the equivalent modulus of the laminates was calculated and the stress curves and curing residual stresses of different regions inside the laminates were further obtained, as shown in Figure 18. For the composite laminates cured in the traditional heating device, affected by the uneven distribution of the temperature field during the heating process, the curing of the internal resins was asynchronous and the inhibitory actions of fibers on the resins were quite different, resulting in obvious differences in the curing residual stresses, corresponding to different measuring points. The curing residual stress of the measuring point 2 with a higher temperature was −28.22 MPa, while that of point 1 with a lower temperature was −20.16 MPa; the difference between the two was 28.56%.

On the other hand, for the laminates cured in the temperature field regulating device, since there was no obvious difference in the curing temperature of each position inside the laminates, the cross-linking reaction of the resins could be carried out simultaneously with the increase of temperature, and the carbon fibers had basically the same inhibitory effect on the thermal expansion and curing shrinkage of the resins. After the experiments, the curing residual stress of the measuring point 1 was −20.89 MPa while that of point 2 was −20.55 MPa; there was only a 1.63% difference between the two. At the same time, compared with the laminates cured in the traditional heating device, the internal maximum curing residual stress was significantly reduced under the condition of uniform microwave heating, falling by 25.97%.

## 5. Conclusions

In this paper, a resonator design scheme that took into account microwave heating uniformity and efficiency was formed, the regulations of different mechanical optimization methods on the uniformity of microwave field were revealed, a regulating device of microwave heating uniformity for composite components was independently developed and, finally, the experimental verification was realized. The analysis showed that the volume of the resonator should be at least greater than 1.59 m^3^, which could have a higher heating efficiency while ensuring the heating uniformity. The optimization and introduction of mechanical structures, such as multi-microwave generators and mode stirrers, would play an important role in improving the uniformity of temperature field and reducing the curing deformation of composites. The maximum temperature difference inside the composite laminates formed by the regulating device established in this paper was only 3.6 °C, which was 79.31% lower than the maximum temperature difference of 17.4 °C in the curing process of the traditional heating device. The curing degrees of the specimens sampled from different positions were essentially the same and the maximum curing degree difference between different samples was only 0.77%. The maximum residual stress was reduced from −28.22 MPa to −20.89 MPa compared with the laminates prepared under traditional microwave heating conditions, a decrease of 25.97%. On the basis of this paper, future work will focus on comprehensively considering the effects of the curing reaction exothermicity, thermal exchange between components and molds on the uniformity of the temperature field of microwave curing, in order to further improve the accuracy and the scope of the application of research.

## Figures and Tables

**Figure 1 polymers-14-03484-f001:**
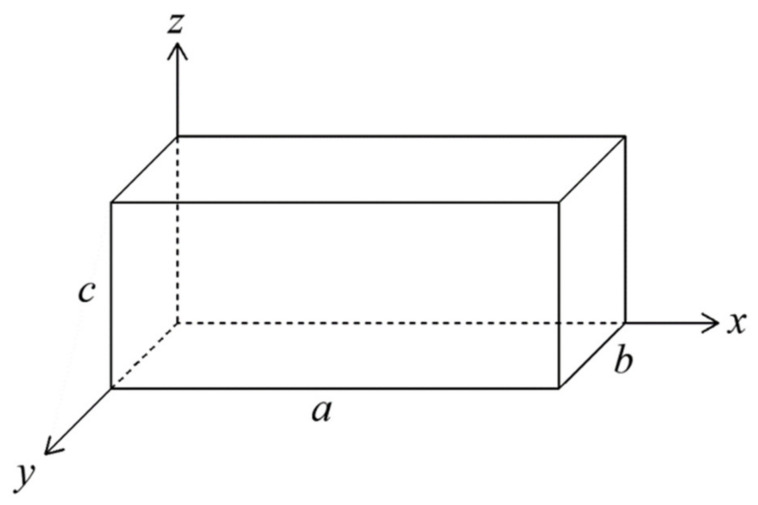
Schematic diagram of the rectangular microwave resonator.

**Figure 2 polymers-14-03484-f002:**
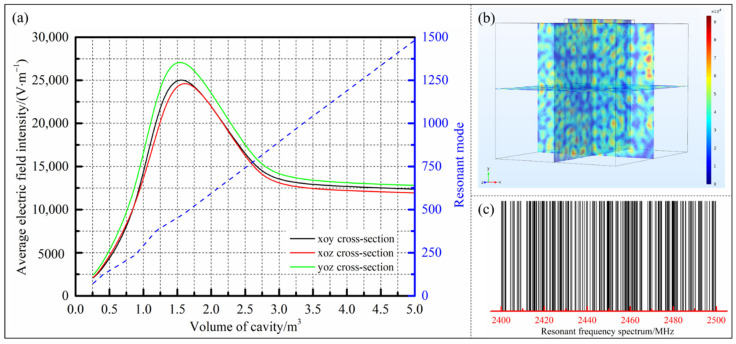
(**a**) The relationship curves of the resonant modes and the average electric field intensity with the volume of cavity; (**b**) the electric field distribution under the critical volume; (**c**) the existing resonant frequency spectrum in the cavity under the critical volume.

**Figure 3 polymers-14-03484-f003:**
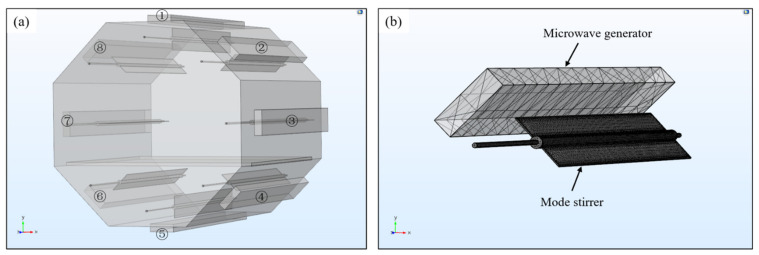
(**a**) Geometric model of the microwave cavity; (**b**) mesh generation of the microwave generator and the mode stirrer.

**Figure 4 polymers-14-03484-f004:**
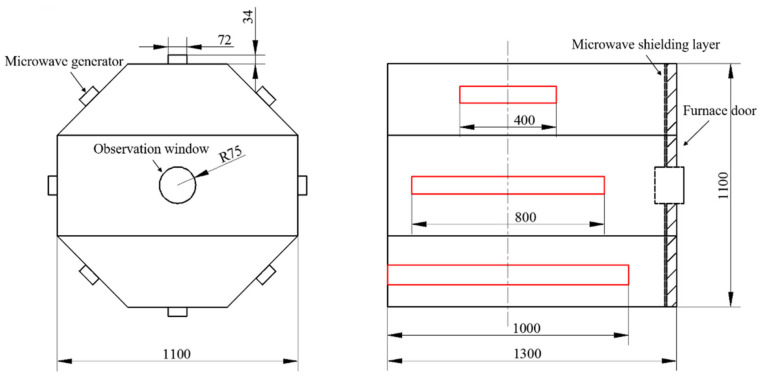
Arrangement of the microwave generators.

**Figure 5 polymers-14-03484-f005:**
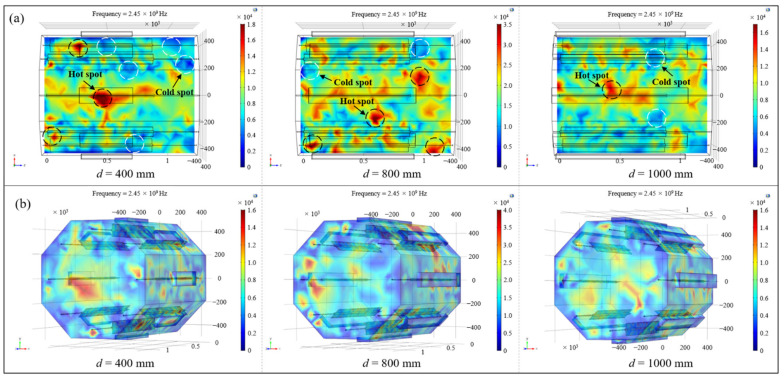
(**a**) The electric field nephograms of the platform; (**b**) the spatial electric field distribution in the cavity.

**Figure 6 polymers-14-03484-f006:**
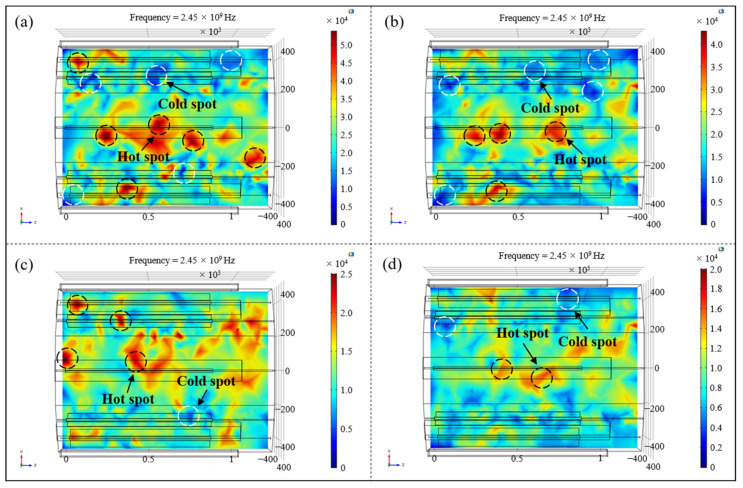
The electric field nephograms on the loading platform of different working conditions: (**a**) number (6); (**b**) number (9); (**c**) number (16); (**d**) number (23).

**Figure 7 polymers-14-03484-f007:**
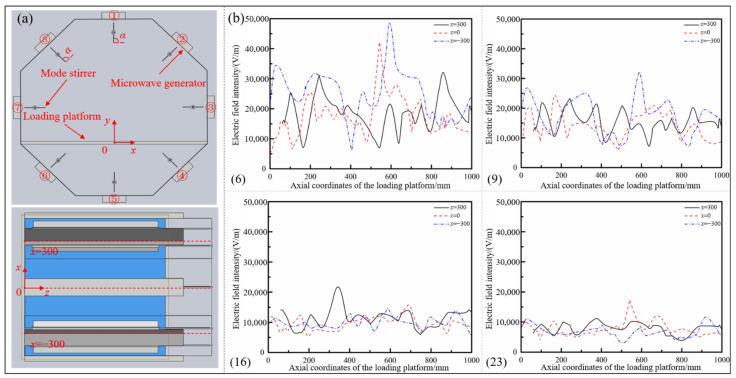
(**a**) Establishment of the associated coordinate system; (**b**) electric field intensity curves under different microwave generators working conditions.

**Figure 8 polymers-14-03484-f008:**
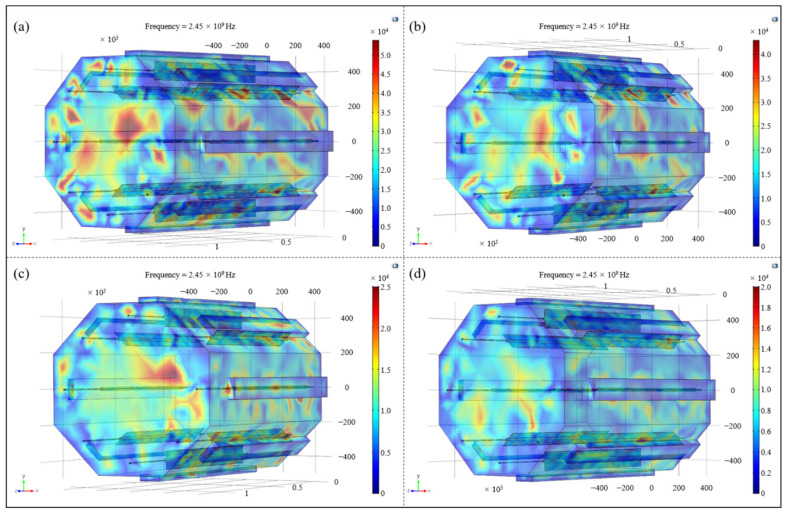
The spatial electric field nephograms under different arrangements of microwave generators: (**a**) number (6); (**b**) number (9); (**c**) number (16); (**d**) number (23).

**Figure 9 polymers-14-03484-f009:**
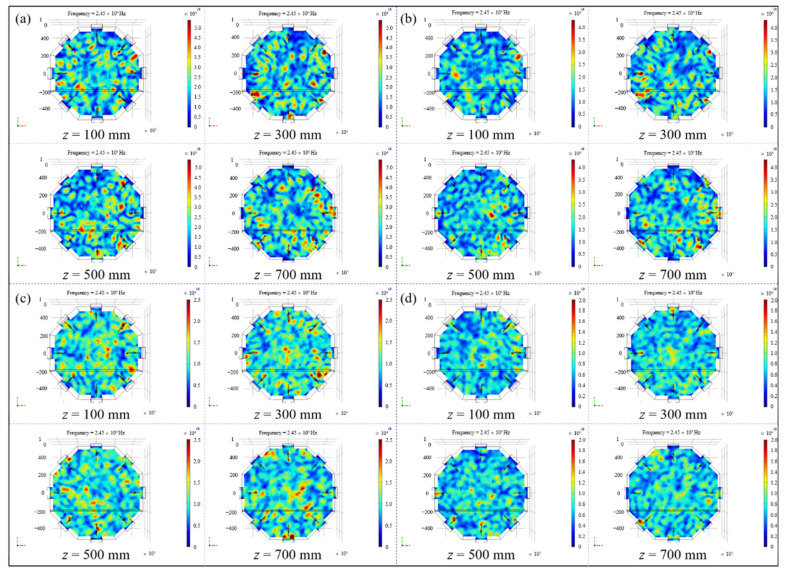
Sliced nephograms of spatial electric field under different arrangements of microwave generators: (**a**) number (6); (**b**) number (9); (**c**) number (16); (**d**) number (23).

**Figure 10 polymers-14-03484-f010:**
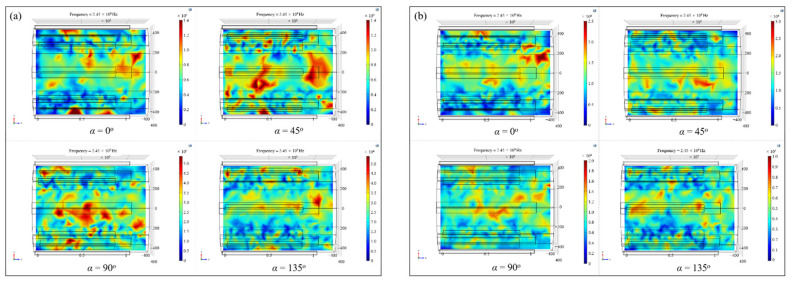
Electric field nephograms on the loading platform during the rotation of mode stirrers under two types of working conditions: (**a**) number (6); (**b**) number (23).

**Figure 11 polymers-14-03484-f011:**
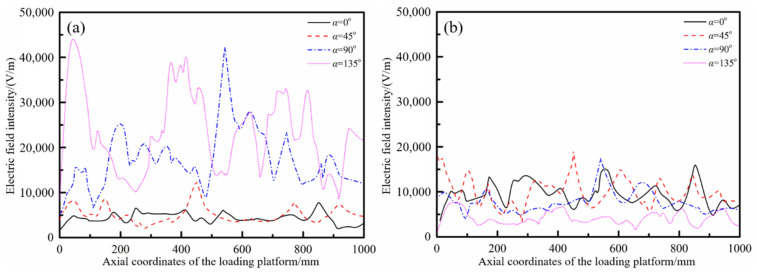
Variations of the electric field intensity along the axis under two types of working conditions: (**a**) number (6); (**b**) number (23).

**Figure 12 polymers-14-03484-f012:**
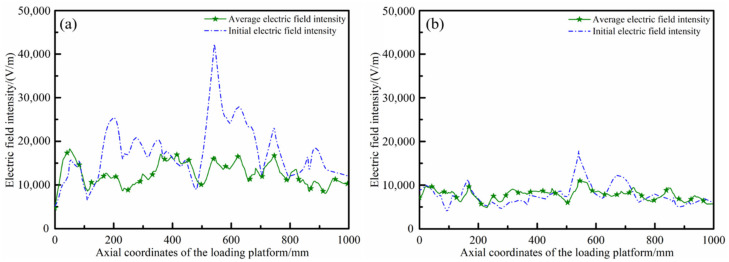
Electric field curves before and after the introduction of the mode stirrers under two types of working conditions: (**a**) number (6); (**b**) number (23).

**Figure 13 polymers-14-03484-f013:**
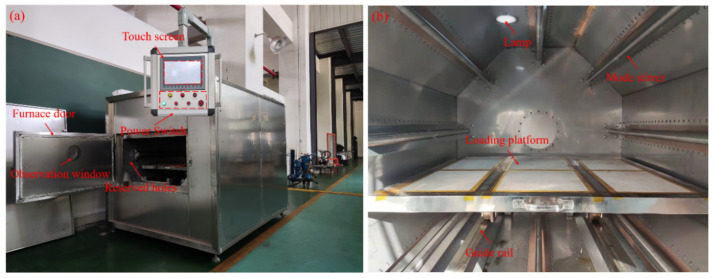
Regulating device of microwave heating uniformity: (**a**) exterior of the device; (**b**) internal structure of the device.

**Figure 14 polymers-14-03484-f014:**
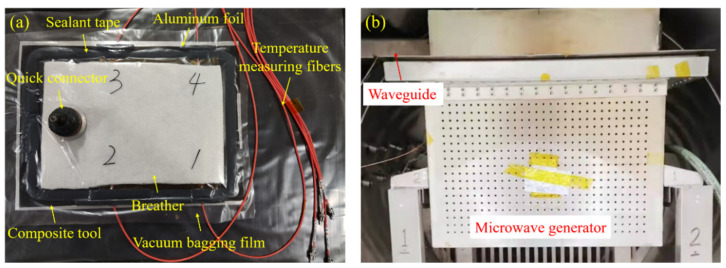
(**a**) Arrangement of the temperature measuring fibers; (**b**) traditional microwave heating platform.

**Figure 15 polymers-14-03484-f015:**
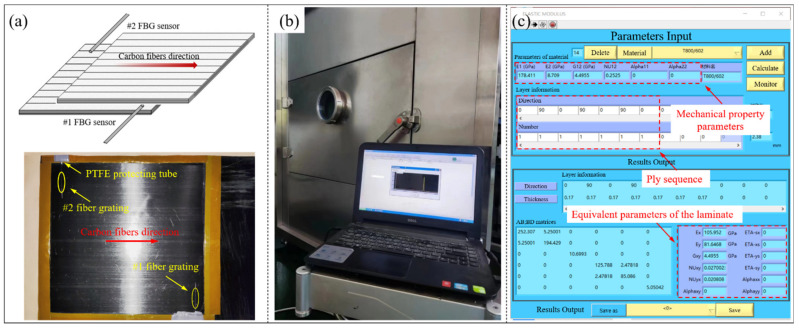
(**a**) Arrangement of the FBG sensors; (**b**) monitoring process during microwave heating of composite laminates; (**c**) equivalent parameters calculation software.

**Figure 16 polymers-14-03484-f016:**
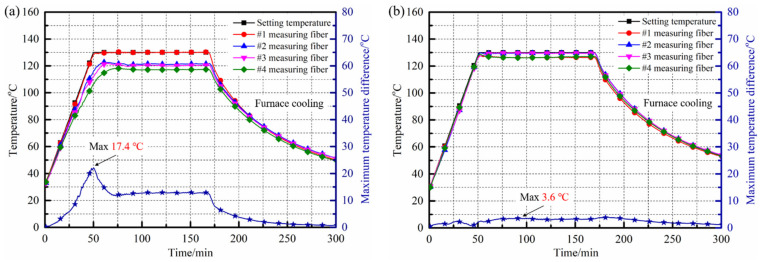
Internal temperature distribution of composite laminates during microwave curing of different devices: (**a**) traditional heating device; (**b**) regulating device.

**Figure 17 polymers-14-03484-f017:**
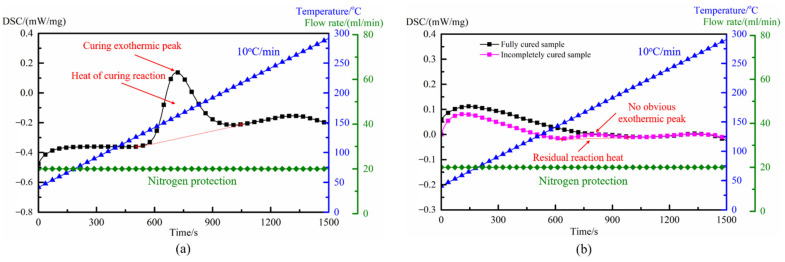
DSC curves: (**a**) uncured prepregs; (**b**) samples after microwave cured.

**Figure 18 polymers-14-03484-f018:**
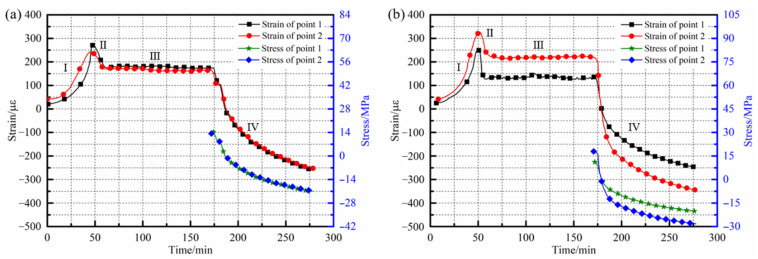
Strain/stress curves of composite laminates during microwave heating process: (**a**) traditional heating condition; (**b**) regulating device heating.

**Table 1 polymers-14-03484-t001:** Setting methods of the microwave generators.

Number	Quantity of Generators	Working Locations	Number	Quantity of Generators	Working Locations
(1)	1	1	(13)	4	2/3/7/8
(2)	2	1/5	(14)	4	2/4/6/8
(3)	2	2/8	(15)	5	1/2/4/6/8
(4)	2	3/7	(16)	5	1/2/3/7/8
(5)	2	4/6	(17)	5	3/4/5/6/7
(6)	2	4/8	(18)	6	1/2/3/5/7/8
(7)	3	1/2/8	(19)	6	1/2/4/5/6/8
(8)	3	1/3/7	(20)	6	1/3/4/5/6/7
(9)	3	4/5/6	(21)	6	2/3/4/6/7/8
(10)	4	1/2/5/8	(22)	7	1/2/3/4/6/7/8
(11)	4	1/3/5/7	(23)	7	2/3/4/5/6/7/8
(12)	4	1/4/5/6	(24)	8	1/2/3/4/5/6/7/8

**Table 2 polymers-14-03484-t002:** Electric field distribution under different arrangement of microwave generators.

Number	Working Locations	CoV	Number	Working Locations	CoV
(1)	1	0.405	(13)	2/3/7/8	0.318
(2)	1/5	0.358	(14)	2/4/6/8	0.335
(3)	2/8	0.397	(15)	1/2/4/6/8	0.353
(4)	3/7	0.351	(16)	1/2/3/7/8	0.335
(5)	4/6	0.394	(17)	3/4/5/6/7	0.338
(6)	4/8	0.407	(18)	1/2/3/5/7/8	0.357
(7)	1/2/8	0.399	(19)	1/2/4/5/6/8	0.328
(8)	1/3/7	0.335	(20)	1/3/4/5/6/7	0.314
(9)	4/5/6	0.379	(21)	2/3/4/6/7/8	0.307
(10)	1/2/5/8	0.389	(22)	1/2/3/4/6/7/8	0.373
(11)	1/3/5/7	0.314	(23)	2/3/4/5/6/7/8	0.304
(12)	1/4/5/6	0.326	(24)	1/2/3/4/5/6/7/8	0.328

**Table 3 polymers-14-03484-t003:** Curing degree of different samples.

Heating Mode	Sample Position	Weight/mg	Residual Heat/(J·g^−1^)	Total Reaction Heat/(J·g^−1^)	Curing Degree
Traditional heating device	1	4.07	0.13	77.32	99.83%
	2	4.38	4.45	77.32	94.24%
	3	6.84	3.91	77.32	94.96%
	4	3.77	1.99	77.32	97.43%
Regulating device	1	3.84	0.72	77.32	99.07%
	2	4.37	0.17	77.32	99.78%
	3	4.43	0.29	77.32	99.62%
	4	4.96	0.77	77.32	99.01%

**Table 4 polymers-14-03484-t004:** Monitoring results with different heating conditions.

Heating Mode		Maximum Strain/µε	Insulation Strain (µε)	Residual Strain/µε
Traditional heating device	Point 1	251.97	135.71	−246.87
	Point 2	326.92	220.06	−345.69
Regulating device	Point 1	273.01	180.71	−255.84
	Point 2	242.82	165.18	−251.69

## Data Availability

The data presented in this study are available on request from the corresponding author.

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
