# Peer review of "Finite Element Simulation and Experimental Research on Uniformity Regulation of Microwave Heating of Composite Materials"

_polymers, 2022, doi:10.3390/polym14173484_

Round 1
Reviewer 1 Report
Dear authors, I congratulate you for conducting the present study.
The Title looks fine but it would be advisable to add the study design in it (if the authors feel it is appropriate)
Abstract look fine.
I suggest the authors to present the keywords by alphabetic order.
The Introduction is very extensive. And the authors lose themselves in a debate which should be shorter and in order to make the reader familiar with the topic of the study. Therefore, although very complete, I recommend the Introduction to be shorter.
The aim sentence is missing.
A clear “Material and Methods” subheading is missing. As mentioned before, the extensive literature review makes the experimental work to lose weight. Try to follow the traditional mode with the major headings “Introduction + Material and Methods + Results + Discussion + Conclusions” independently of the subheading that may be inside each major heading. This format will help the reader to follow the study.
I suggest the “Results and Discussion” to be divided in “Results” and “Discussion” headings.
Once again the Conclusion are too long. Can it be shorten?
I would feel appropriate to debate limitations and strength of the study and possible future study recommendations.
Author Response
Dear Reviewer 1:
Thank you for your letter and for the comments concerning our manuscript entitled “Simulation and experimental research on uniformity regulation of microwave heating of composite materials” (ID: Polymers-1864770). We accept the suggestions from the reviewer and those comments are valuable and very helpful for revising and improving our paper. After receiving comments, all co-authors discussed and engaged to revising manuscript according to the comments. All the revisions were made as follows concretely, please see the attachment.
Thank you again.
Warmest regards
Chenglong Guan

Reviewer 2 Report
In the abstract, there is no clear information regarding the sort of composites tested.
All of the equations require appropriate references.
There should be a gap between the numerical number and the units.
Most of the figures, particularly figure 2b, should have higher resolution.
DSC method information should be supplied.
Author Response
Dear Reviewer 2:
Thank you for your letter and for the comments concerning our manuscript entitled “Simulation and experimental research on uniformity regulation of microwave heating of composite materials” (ID: Polymers-1864770). We accept the suggestions from the reviewer and those comments are valuable and very helpful for revising and improving our paper. After receiving comments, all co-authors discussed and engaged to revising manuscript according to the comments. All the revisions were made as follows concretely, please see the attachment.
Thank you again.
Warmest regards
Chenglong Guan

Reviewer 3 Report
The manuscript "Simulation and experimental research on uniformity regulation 2 of microwave heating" submitted by Guan, et al. t Polymers provides a very nice combination of theoretical modelling and experimental verification on the application of microwave heating towards the curing of composites. The FEA modelling appears to be thorough. The authors did not include any term on the curing reaction exothermicity, which can exacerbate the non-uniform heating effect (in cases of an exothermic curing reaction) but which can conversely provide a damping effect on this for an endothermic reaction. The authors might make a comment on this in their manuscript.
The experimental verification shows a significant increase in curing uniformity on addition of mode stirrers to the curing process. This result could be of significant interest if this finds general application.
This manuscript is ready for publication.
Author Response
Dear Reviewer 3:
Thank you for your letter and for the comments concerning our manuscript entitled “Simulation and experimental research on uniformity regulation of microwave heating of composite materials” (ID: Polymers-1864770). We accept the suggestions from the reviewer and those comments are valuable and very helpful for revising and improving our paper. After receiving comments, all co-authors discussed and engaged to revising manuscript according to the comments. All the revisions were made as follows concretely, please see the attachment.
Thank you again.
Warmest regards
Chenglong Guan

Round 2
Reviewer 1 Report
Dear authors, I have no more concerns although I recognize that you did not have into consideration the corrections I suggested. In the end of the day, the work, and the authors, is the one that loses by not following the usual scientific work format.